# Assessing white matter plasticity in a randomized controlled trial of early literacy training in preschoolers

Sendy Caffarra[1,2]*, Iliana I. Karipidis[3,4,5], John Kruper[6,7], Emily Kubota[8],
Adam Richie-Halford[2], Megumi Takada[9], Ariel Rokem[6,7], Jason D. Yeatman[2,9]

1 Department of Biomedical, Metabolic and Neural Sciences, University of Modena and Reggio Emilia, Modena, Italy, 2 Division of Developmental-Behavioral Pediatrics, Stanford School of Medicine, Stanford, California, United States of America, 3 Department of Child and Adolescent Psychiatry and Psychotherapy, University Hospital of Psychiatry Zurich, Zurich, Switzerland, 4 Neuroscience Center Zurich, University of Zurich and ETH, Zurich, Switzerland, 5 Department of Psychiatry and Behavioral Sciences, Stanford School of Medicine, Stanford, California, United States of America, 6 Department of Psychology, University of Washington, Seattle, Washington, United States of America, 7 eScience Institute, University of Washington, Seattle, Washington, United States of America, 8 Department of Psychology, Stanford University, Stanford, California, United States of America, 9 Graduate School of Education, Stanford University, Stanford, California, United States of America

* caffarra@unimore.it

## Abstract

Reading is a cognitive skill that requires our brain to go through a myriad of changes during learning. While many studies have described how reading acquisition shapes children's brain function, less is known about the impact of reading on brain structure. Here we examined short-term causal effects of reading training on preschoolers' behavior and white matter structure. Forty-eight English-speaking preschoolers (4y10m to 6y2m) participated in a randomized controlled trial where they were randomly assigned to two training programs: the Letter training program was focused on key skills for reading (e.g., decoding and letter knowledge), while the Language training program strengthened oral language comprehension skills without exposure to text. Longitudinal behavioral data showed that only the Letter Training group increased letter knowledge and decoding skills after the two-week training. Diffusion MRI measures (FA and MD) of eighteen white matter pathways (including the left arcuate and the left inferior longitudinal fasciculus) did not reveal any statistically significant changes for either group despite high degrees of scan-rescan reliability across sessions. These findings suggest that a two-week reading training program can cause changes in preschoolers' letter knowledge and decoding abilities, without being accompanied by measurable changes in the diffusion properties of the major white matter pathways of the reading network. We conclude highlighting possible constraints (i.e., age, training onset and duration, cognitive profile) to reading-related white matter plasticity.

## Introduction

Reading is a complex cognitive skill that has an impact on brain structure and function. Learning to decode written language not only changes the way the brain functions [1,2], but is

**Data availability statement:** BIDS MRI dataset is available on OpenNeuro. doi:10.18112/openneuro.ds005572.v1.0.0

**Funding:** SC conducted this research as part of the projects FAR Mission Oriented 2022 and PRIN PNRR 2022 P2022SMEJW, which were funded by the European Union – Next Generation EU. IK was supported by the Stanford Maternal and Child Health Research Institute award. AR and JK were funded by NSF grant 1934292 and NIH grants RF1 MH121868 and R01EB027585. JK was additionally supported through the NSF Graduate Research Fellowship DGE-2140004, NIH grants R21HD092771 and R01HD095861. The funders had no role in study design, data collection and analysis, decision to publish, or preparation of the manuscript.

**Competing interests:** The authors have declared that no competing interests exist.

also associated with changes in the structural properties of white matter pathways [3–5]. However, it is still unclear under which conditions (e.g., quantity and quality of the training, developmental stage), and at what time scale experience-dependent structural changes emerge. In addition, the relationship between learning-driven changes in reading behavior and brain plasticity is still underspecified. To deepen our understanding of the short-term effects of the initial phase of reading acquisition on brain structure, we ran a diffusion magnetic resonance study (dMRI) that used a randomized controlled trial in preschoolers to test how training in letter-speech sound knowledge affects behavior, white matter structure, and their relationship over the course of two weeks.

A growing body of research reports a relationship between reading experience and structural properties of white matter pathways [6]. Among the white matter tracts of the reading circuitry there are the left arcuate (AF, [7]), the left inferior longitudinal fasciculus (ILF, [8]), the inferior fronto-occipital fasciculus (IFOF, [9]), and superior longitudinal fasciculus (SLF, [10]). Diffusion properties of these white matter tracts, such as fractional anisotropy (FA) and mean diffusivity (MD), have been related to reading performance in a single time point (concurrent or prior to the dMRI acquisition, [11–15]). Longitudinal findings have confirmed a link between reading development and changes in white matter microstructural properties, especially for the left AF and the left ILF [4,5,16,17]. Recent studies have started to highlight the presence of a dynamic relationship between the longitudinal trajectories of white matter structure and changes of reading performance over time [18,19]. For instance, studies focused on long-term reading-related structural plasticity reported that typically developing children show increased FA and/or decreased MD in the left AF and the left ILF as reading scores improve. These structural changes are evident over one to four years of formal reading instruction [15,17,20–22]. Moreover, the rate of FA change in the left AF relates to the rate of change in reading performances over a period of five years of reading instruction [19].

Longitudinal studies focused on a shorter period of reading training have led to mixed findings. Table 1 summarizes the available longitudinal research on reading-dependent changes of white matter diffusion properties. Unfortunately, the picture provided by these studies is only partial since most research so far has been focused on reading interventions for children with (or at-risk of) reading disorders and its short-term effects on white matter properties.

Although the number of studies conducted on this topic is still low, some preliminary trends can be highlighted given the available findings. First, all studies consistently showed that only children who received reading intervention improved their reading performance, confirming the efficacy of short-term reading training programs [4,5,16,23,24]. Second, these intervention-specific behavioral changes were not always accompanied by fast learning-driven white matter changes, suggesting that behavioral and brain structure changes do not necessarily co-occur or this co-occurrence might be specific to a subset of white matter diffusion properties [3–5,23–25]. Third, there is scarce and mixed evidence on how effects generalize across different age ranges and types of intervention program, pointing to the need for additional research on short-term coupling between reading behavior and white matter plasticity [4,5,16]. The minimal duration of intervention associated with structural changes is also unclear. While a few studies reported structural neuroplasticity after a hundred of hours of intervention [5,16], others have shown white matter changes within the first 50 hours of intervention [3,4,23]. Finally, all studies listed above focused on remediation programs [4,16]. Hence, they provide insights on rapid white matter changes that might also reflect compensatory mechanisms of the reading circuitry, or other factors that are unique to older children with dyslexia, rather than solely plasticity due to the experience of learning to read.

**Table 1. Studies on white matter changes due to short-term reading intervention programs.**

| Paper | Intervention | | Age (y) | Sample Size | Language | Intervention specific effects (Group x Session) | | | |
|---|---|---|---|---|---|---|---|---|---|
| | Duration (h) | Type | | | | Behavior | dMRI | Behavior - dMRI coupling | Diffusion property examined |
| Keller et al. 2009 [5] | 100 | g-p | 8-10 | 72 (37 C) | English | x | x (FA, RD) | x (FA, RD) | FA, RD, AD |
| Huber et al. 2018 [4] | 160[a] | i-p | 7-12 | 43 (19 C) | English | x | x (FA, MD) | x (MD) | FA, MD |
| Huber et al. 2021 [23] | 160[a] | i-p | 7-12 | 73 (41 C) | English | x | x (MD, MDe) | N.A. | MD, MDe, AWF, DK, R1 |
| Partanen et al. 2021 [24] | 24[b] | g-p | 8-9 | 35 (22 C) | English | x | – | N. A. | FA, MD |
| Economou et al. 2022 [25] | 18 | i-c | 5-6 | 83 (52 C) | Dutch | N.A. | – | N.A. | FA |
| Economou et al. 2023 [3] | 18 | i-c | 5-6 | 90 (59 C) | Dutch | N.A. | x (MWF) | N.A. | MWF |
| Meisler et al. 2024 [16] | 100-120 | g-p | 7-9 | 41 (15 C) | English | x | N.A. | x[c] (FA, MD) | FA, MD |

Intervention Type: g = intervention was carried out in small groups; i = intervention was carried out individually; c = intervention was computerized; p = intervention was carried out in person.

Sample Size: overall sample size including the experimental and the control groups. C = control group sample size. The experimental groups always included children with reading disorders or at-risk of reading disorders who were enrolled in intensive remediation programs. The control groups included typically developing children, as well as children with or at-risk of reading disorders. They either did not receive any intervention or were enrolled in a non-language specific training program.

Intervention specific effects: x: present; -: absent, N.A.: not available. FA = fractional anisotropy; RD = radial diffusivity; AD = axial diffusivity; MD = mean diffusivity; MDe = extra-axonal mean diffusivity; AWF = axonal water fraction; DK = diffusion kurtosis; T1rt = T1 relaxation time; MWF = myelin water fraction.

[a]The dMRI effects were visible already after 46 hours of intervention.[b] 225 hours only for 4 kids. [c] It would not survive a multiple comparison correction.

The present exploratory study aims to complement the available research on short-term white matter plasticity by focusing on language and literacy training programs in typically developing preschool children. A randomized controlled trial was conducted with English-speaking preschoolers who were enrolled in a two-week program which either trained reading or spoken language skills (Letter and Language program, respectively). Behavioral and dMRI measures were collected before and after the training. To better characterize the quality and consistency of children's dMRI measures over time, scan-rescan reliability metrics were calculated for each dMRI measure (FA and MD) and white matter tract. We expected to observe behavioral changes in reading performances only in the group enrolled in the Letter program. To test whether structural neuroplasticity observed in reading intervention can be generalized to typically developing preschool children, we compared the structural properties of 18 white matter tracts before and after training, including those belonging to the reading brain circuitry (e.g., left AF, left ILF, left IFOF, left SLF).

## Materials and Methods

### Participants

Forty-eight English-speaking preschoolers (5 years of age; range: 58-74 months) participated in a randomized controlled trial in the summer before starting kindergarten (Fig 1).

The recruitment period started on June 2nd and ended on November 19th, 2019. An initial behavioral session ensured that all children participating in the study satisfied the following inclusion criteria: not knowing all uppercase letters and their corresponding sounds; having a Peabody Picture Vocabulary Test raw score higher than 85 (PPVT, 4th Edition; [26]); having normal or corrected to normal vision; being able to hold still for 5 minutes during an MRI mock scan. Table 2 summarizes the demographic characteristics and behavioral measures

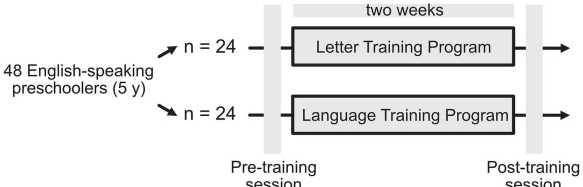

**Fig 1. Graphical representation of the randomized controlled trial.**

Table 2. Overview of the characteristics of the Letter and Language groups before training. Average scores are reported followed by standard deviations within parentheses. Raw scores are reported.

|  | Letter Training | Language Training | X2 | p |
|---|---|---|---|---|
| N | 24 | 24 |  |  |
| N of females | 16 | 13 | 0.260 | 0.390 |
|  | Letter Training | Language Training | t | p |
| Age in months | 62.0 (3.2) | 63.8 (3.7) | 1.825 | 0.075 |
| Pre MRI-camp time difference (days) | 18.3 (5.2) | 22.2 (8.6) | 1.660 | 0.104 |
| Post MRI-camp time difference (days) | 6.2 (4.0) | 7.3 (7.6) | 0.594 | 0.555 |
| Alphabet Knowledge | 64.3 (26.1) | 67.2 (23.5) | 0.567 | 0.572 |
| Decoding | 2.0 (4.1) | 1.8 (4.0) | 0.276 | 0.783 |
| Phoneme Matching | 10.1 (2.4) | 10.5 (2.7) | 0.865 | 0.389 |
| Phoneme Segmenting | 1.5 (1.3) | 1.9 (2.2) | 1.131 | 0.261 |
| Expressive Vocabulary | 83.2 (12.1) | 85.9 (13.6) | 1.029 | 0.306 |
| Story Complexity | 11.8 (5.3) | 13.0 (4.8) | 1.203 | 0.232 |
| Story Grammar | 9.3 (3.3) | 9.8 (2.9) | 0.815 | 0.417 |

(the description of these tests are reported in the "Behavioral data acquisition and analyses" section) of the two groups of children before training. No between-group differences were present prior to training. No neuropsychological or psychiatric disorder was reported. All children gave their assent to participate in the study, and their parents (or legal guardians) signed an informed consent form. The study was approved by the Institutional Review Board of the University of Washington.

## Procedure

Participants were randomly assigned to one of two different training programs: a Letter (n = 24) or a Language (n = 24) program, which were organized into a fun and engaging summer camp. The Letter program followed the Slingerland method [27] and was focused on the foundational skills of reading such as letter recognition, letter-speech sound associations, and phonemic awareness (e.g., blending and segmentation of syllables and trigrams; Table 3). The Language program focused on oral linguistic abilities such as recognizing syntactic categories in spoken sentences, listening, comprehending and retelling stories, learning new vocabulary (Table 3). Critically, the Language program did not include any exposure to written language compared to the Letter program which was almost exclusively focused on written language. Each training program was delivered to a small group of children (n = 6) by three teachers, who had a background in Education or Speech pathology. Each program lasted two weeks (3 hours/day, 5 days/week, 30 total hours) and was based on pedagogical models of Direct

**Table 3. Description of the daily activities performed in the Letter and Language Training programs.**

| Letter Training Program daily activities | |
|---|---|
| Free game (20 min) | Children could freely play with blocks, puzzles, and modeling clay. |
| Phonological awareness (25 min) | Children were introduced to segmentation and blending of syllables, trigrams (C-VC and CVC) and onset-rime words through songs and interactive games. |
| Letters (25 min) | Children learned two lowercase letters per day based on letter-picture correspondences and whiteboard writing activities. |
| Blending and Decoding (25 min) | Children were guided to blend three letters together and then decide whether the outcome was a real word or not. |
| Center activities (20 min x 2 daily sessions) | Children rotated among four different learning stations to reinforce what was learned in the daily session. |
| Story time (10 min) | The teachers read a brief (5-10 minutes) story to the kids. |
| **Language Training Program daily activities** | |
| Free game (20 min) | Children could freely play with blocks, puzzles, and modeling clay. |
| Syntax awareness (25 min) | Children built sentences using a set of picture cards, which represented different words. They were instructed about the function of different words in a sentence. Each card was color coded based on the grammatical category the word belonged to (e.g., noun, verb, adjective), |
| Listening and comprehension (25 min) | Children listened to a story and were guided to identify different narrative elements (e.g., characters, theme, problem). |
| Vocabulary (25 min) | Children were introduced to the meaning of new words based on picture cards, context-based information, personal experience, and examples. Simple exercises were proposed where kids had to use the new words in the right context. |
| Center activities (20 min x 2 daily sessions) | Children rotated among four different learning stations to reinforce what was learned in the daily session. |
| Story time (10 min) | The teachers read a brief (5-10 minutes) story to the kids, who were then asked to identify story elements based on what they learned in the listening and comprehension daily activity. |

Instruction and Gradual Release of Responsibility. Letter and Language activities adopted a multisensory approach involving vision, audition and kinesthetics. Both programs had the same daily schedule and the learning process was scaffolded, so that the content of the activities followed an increasing degree of complexity. Each daily session started with 20 minutes of free play and ended with story time (Table 3). To maximize consistency of the training delivery over time and across participants the same group of teachers administered both Letter and Language activities. In addition, two pilot camps were run before the experiment to reach high levels of coherence among teachers' styles. During these pilot camps of four days the three teachers could practice and coordinate to minimize differences in the way each activity was delivered.

Behavioral and diffusion magnetic resonance imaging (dMRI) measures were collected before and after each training program.

## Behavioral data acquisition and analyses

During the behavioral session, alphabet knowledge was assessed and a series of standardized tests was administered.

Alphabet knowledge was tested through flashcards presented in random order. There were 26 flashcards for uppercase letters, and 26 flashcards for lowercase letters. Each of the 26 cards was shown to the child and they were asked "What letter is this?" and "What sound does it make?". The total raw score of the alphabet knowledge test was 52, both for upper and lowercase letters.

Decoding skills were assessed using the Pseudoword decoding list from the "PALS 1-3:Phonological awareness literacy" [28]. During this task, children were required to read a

list of 20 CVC pseudowords at their own pace. The decoding raw score corresponded to the number of pseudowords correctly read (total raw score: 20).

Phoneme Matching skills were assessed using the Initial and Final Sound Matching subtests from the "Phonological and print awareness scale" [29]. Children were required to isolate and match the first or last phoneme in 18 pairs of words. The total raw score corresponded to the total number of correct responses.

Phoneme segmentation skills were assessed using the Phonemic Awareness subtest from "Phonological and print awareness scale" [29]. Kids were required to segment individual phonemes in 12 real words of one or two syllables. The total raw score corresponded to the number of correct responses.

Productive vocabulary was measured through the Expressive Vocabulary Test (Third Edition [30]). This is a picture naming task with 190 test items in order of increasing difficulty where naming accuracy is assessed.

Children's narrative skills were assessed using the Test of Narrative Retell subtest from "The Narrative Language Measures" [31]. Children's story retellings were evaluated based on two dimensions (according to [31]): language complexity and story grammar. The former subscale evaluated the complexity of the produced sentence structures on 0-2 or 0–3 point ratings (e.g., average sentence length, presence of causal connections, temporal and adversative conjunctions, temporal subordinate clauses, adverbs, low frequency words). Story grammar was evaluated on 0-2 or 0–3 point ratings based on the presence of structural elements in children's story retelling, such as the description of a setting, a problem, an attempt to solve the problem and its consequences. Raw scores of each behavioral measure were used in statistical analyses.

Two-tailed t-tests showed that Letter and Language groups did not differ in any behavioral measure prior to training (software and module used: python v3.11, scipy.stats v1.15.0, Table 2). For each behavioral measure, a linear mixed effect model (LME) was run to test for training effects. Time (pre vs post session), Training Type (Letter vs Language) and their interaction were included as fixed effects (software and module used: python v3.11, statsmodels v0.14.4). By-subject random intercepts were included in the models. Random slopes were also included as they improved model fit (average decrease of BIC after adding random slopes: 7.1).

### dMRI data acquisition, preprocessing and analyses

MRI data was collected through a 3 T Phillips Achieva scanner with a 32-channel head coil (Philips, Eindhoven, Netherlands). A whole-brain anatomical volume at 1.0 x 1.0 x 1.0 mm resolution was acquired using a T1-weighted MPRAGE sequence (TR 9.2 s, TE 4.35 ms, matrix size 224 x 224, field of view 224 x 224 x 170, 170 slices). Diffusion-weighted magnetic resonance imaging (dMRI) data of the full brain were acquired with a spatial resolution of 2.0 mm3 (anterior-posterior phase encoding direction). A diffusion-weighted imaging (DWI) scan was acquired with 32 non-collinear directions (b-value = 1500 s/mm2; TR = 7200; TE = 83 ms). Four volumes with no diffusion weighting were also acquired (b-value = 0). To correct for echo-planar imaging distortions, one scan with a reversed phase encoding direction (posterior-anterior) and with three non-diffusion-weighted volumes was collected.

The T1-weighted (T1w) images were corrected for intensity non-uniformity (INU) using N4BiasFieldCorrection [32], ANTs 2.3.1), and used as T1w-reference throughout the workflow. The T1w-reference was then skull-stripped using antsBrainExtraction.sh (ANTs 2.3.1), using OASIS as target template [33]. Spatial normalization to the ICBM 152 Nonlinear Asymmetrical template version 2009c [34] was performed through nonlinear registration with antsRegistration (ANTs 2.3.1, [35], using brain-extracted versions of both T1w volume and template. Brain tissue segmentation of cerebrospinal fluid (CSF), white-matter (WM) and gray-matter (GM) was performed on the brain-extracted T1w using FAST (FSL 6.0.3, [36]).

DMRI preprocessing and reconstruction were carried out using QSIprep 0.13.0RC2 ([37–39]), which is based on Nipype 1.6.0[37–39], Nilearn 0.7.1 [40] and Dipy 1.4.0 [41]. The preprocessing included topup distortion, MP-PCA denoising, motion and Eddy current correction (q-space smoothing factor = 10, 5 iterations; [42–45]). Only experimental sessions with a maximum framewise displacement below 4 mm and an average framewise displacement below 1 mm were further analyzed (Letter group pre-training session: 21; Letter group pre-training session: 22; Language group pre-training session: 19; Language group post-training session: 20). Multi-tissue fiber response functions were estimated using the dhollander algorithm as implemented in MRtrix3 [46]. Fiber orientation distributions (FODs) in each voxel were estimated via constrained spherical deconvolution (CSD,[47,48] using an unsupervised multi-tissue method[49,50]. Anatomically constrained tracking (ACT) was applied. FODs were intensity-normalized using mtnormalize[51]. Probabilistic tractography was carried out using the following QSIprep parameters: 1M streamlines, minimum length: 30 mm, maximum length: 250 mm. Fiber segmentation was carried out using pyAFQ 0.9 default parameters (https://yeatmanlab.github.io/pyAFQ;[52,53] cleaning iterations = 5, distance threshold = 5 SD, length threshold: 4 SD). Eighteen default tracts were segmented: Left/Right Arcuate, Left/Right Anterior Thalamic Radiation, Left/Right Cingulum, Left/Right Corticospinal Tract, Anterior/Posterior Forceps, Left/Right Inferior Fronto-Occipital Fasciculus, Left/Right Inferior Longitudinal Fasciculus, Left/Right Superior Longitudinal Fasciculus, Left/Right Uncinate. Diffusion metrics were calculated using the diffusion tensor model (DTI[54,55] and projected onto the tracts. Each streamline was resampled into a fixed number of nodes (n = 100), and average values of fractional anisotropy (FA), and mean diffusivity (MD) were calculated for each node. FA and MD were mapped onto each tract, weighting the values based on the streamline's distance from the core of the tract [52,56]. These final steps were done to obtain the tract profile, which refers to all FA (or MD) values obtained for each node along the tract of a single participant.

For each white matter tract and diffusion property (FA and MD), we calculated scan-rescan reliability metrics to quantify the consistency of two types of dMRI measures between experimental sessions: profile and subject reliability (as in [53]). Profile reliability was first calculated at a subject-level as the Pearson correlation between the tract profiles of the pre and post-training sessions (software and module used: python v3.11, scipy.stats v1.15.0). These correlation coefficients were finally averaged across participants to obtain the final profile reliability score of each tract. To calculate the subject reliability, we first obtained the median value across the 100 nodes of a single tract and participant (individual tract value). We then calculated the Pearson correlation between the individual tract values of the pre-training session and the individual tract values of the post-training session.

An LME model was run on the average FA and MD values of each tract profile to test for structural changes due to the type of training received. Time (pre vs post session), Training Type (Letter vs Language) and their interaction were included as fixed factors (software and module used: python v3.11, statsmodels v0.14.4). By-subject random intercepts were also included. Random slopes were not included as they did not improve model fit (average increase of BIC after adding random slopes in FA models: 8.2; average increase of BIC after adding random slopes in MD models: 6.6). Similar LME models were fitted for each single node of each tract profile to test for training effects in discrete portions of the tracts. It is important to note that the presence of training-induced changes does not necessarily exclude high reliability scores between the two experimental sessions. High reliability scores are still compatible with general training-induced changes in FA/MD without drastic between-session variations of the relationships across nodes or participants. On the other hand, training-induced effects that are uniform across participants but not uniform along the tracts would correspond to a reduced profile reliability, while keeping subject reliability high.

Finally, we tested for a potential coupling between reading-related behavioral changes and reading white matter tracts. For these follow-up analyses, we focused on those behavioral measures showing a specific effect of Letter training. Pearson correlations were calculated between reading-related behavioral changes and structural changes of left AF and ILF across participants. A Bonferroni correction was applied by adjusting the p values based on the total number of computed correlations (n = 16). The full code for behavioral and dMRI analysis is available at this link https://github.com/SendyCaffarra/PREK-analysis.git

## Results

### Behavioral results

Table 4 summarizes the LME results for each behavioral test. Only models on alphabet knowledge (average accuracy of upper and lowercase) and decoding skills showed a significant interaction between Training Type and Time (alphabet knowledge: $\beta$ = 0.822, SE = 0.370, t = 2.224, p = 0.026, d = 0.726; decoding skills: $\beta$ = 0.970, SE = 0.326, t = 2.972, p = 0.003, d = 0.909), indicating that children participating in the Letter Training improved their letter knowledge ($\beta$ = 2.770, SE = 0.784, t = 3.532, p < 0.001, d = 0.406) and decoding ability ($\beta$ = 1.689, SE = 0.496, t = 3.403, p = 0.001, d = 0.855), while children participating in the Language Training group did not show such behavioral changes (alphabet knowledge: $\beta$ = 0.896, SE = 0.776, t = 1.154, p = 0.248; decoding skills: $\beta$ = 0.194, SE = 0.415, t = 0.469, p = 0.639; Fig 2). Apart from the significant interaction, a main effect of Time was also present (alphabet knowledge: $\beta$ = 1.813, SE = 0.370, t = 4.905, p < 0.001, d = 0.309; decoding skills: $\beta$ = 0.698, SE = 0.326, t = 2.140, p = 0.032, d = 0.381) indicating higher alphabet knowledge and decoding scores in the post-training session, which could possibly reflect additional repeated practice effects. The main effect of Training Type was not significant indicating no overall differences between groups (alphabet knowledge: $\beta$ = 0.385, SE = 1.803, t = 0.213, p = 0.831; decoding skills: $\beta$ = 0.371, SE = 0.491, t = 0.755, p = 0.450)

### dMRI results

**Scan Rescan reliability.** Our dMRI measures showed high degrees of scan-rescan reliability between the two experimental sessions (profile reliability: FA, median r = 0.99, range: 0.93-0.99; MD, median r = 0.92, range: 0.65-0.99; subject reliability: FA, median r = 0.83, range: 0.62-0.90; MD, median r = 0.87, range: 0.55-0.92; Fig 3, for Subject reliability of each experimental group, see S1 and S2 Figs).

**Training effects on dMRI measures.** No structural changes were observed between experimental sessions (FA: all ts < 2; MD: all ts < 2.5) or between the two groups (FA: all ts < 2.8; MD: all ts < 2). Interactions between Training Type and Time were not significant (FA: all ts < 2.12; MD: all ts < 2), suggesting that no statistically significant changes were observed

**Table 4. Summary of behavioral LME results relative to the Training Type x Time interaction.**

|  | β | SE | t | p |
|---|---|---|---|---|
| Alphabet Knowledge* | 0.822 | 0.370 | 2.224 | 0.026 |
| Decoding** | 0.970 | 0.326 | 2.972 | 0.003 |
| Phoneme Matching | 0.344 | 0.267 | 1.289 | 0.197 |
| Phoneme Segmenting | 0.080 | 0.113 | 0.707 | 0.480 |
| Expressive Vocabulary | 0.076 | 0.716 | 0.106 | 0.916 |
| Story Complexity | 0.494 | 0.526 | 0.940 | 0.347 |
| Story Grammar | 0.222 | 0.289 | 0.768 | 0.442 |

# Behavioral results

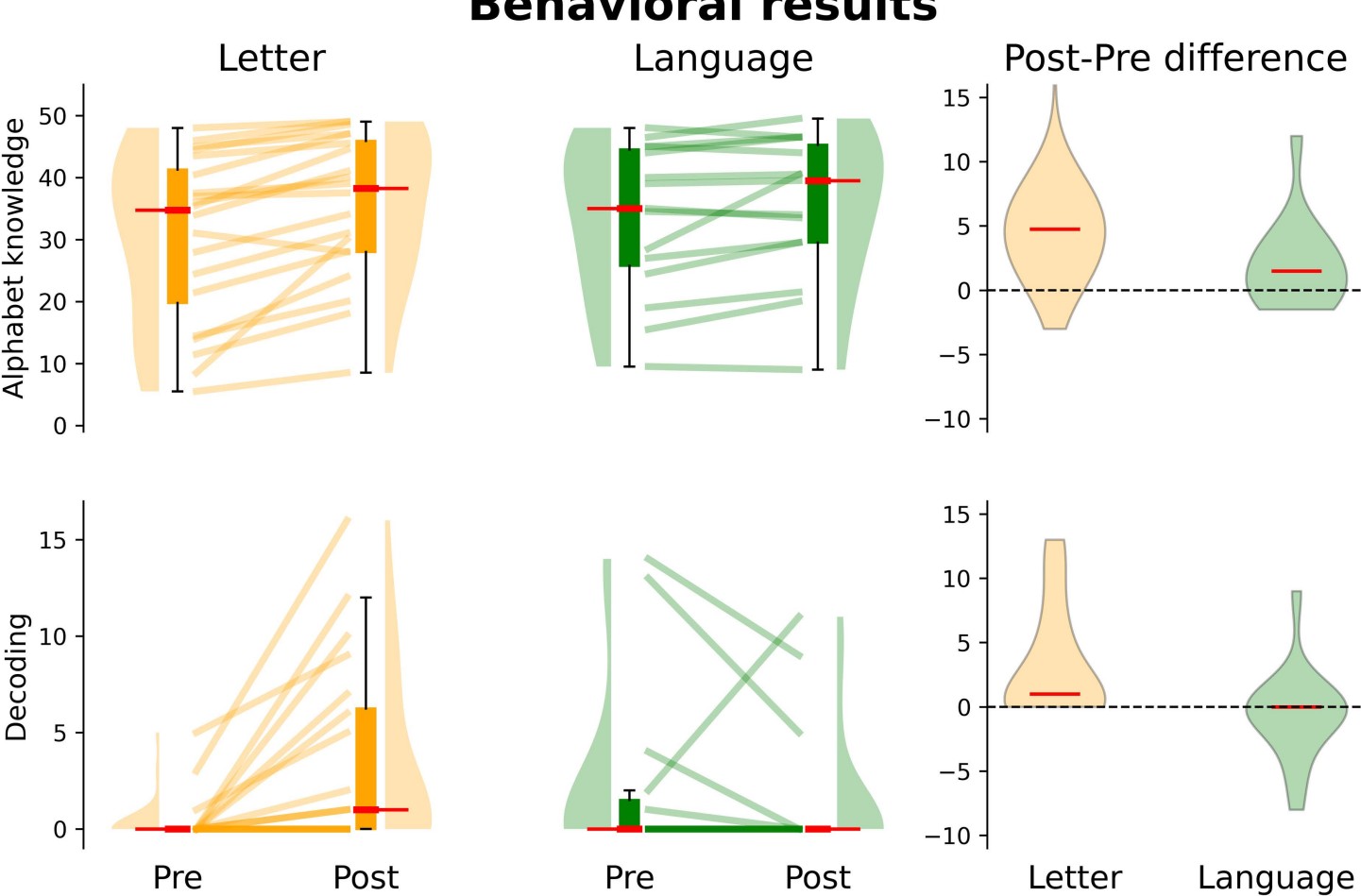

**Fig 2. Training-related behavioral changes.** First two columns: behavioral changes in alphabet knowledge from the Letter and Language Training groups for each experimental session. The third column shows the distribution of alphabet knowledge changes (i.e., difference between the individual scores obtained in the post and pre-training sessions) for each group.

for either group over the 2-week training period (Fig 4 shows the results for the left arcuate; for similar results on the left ILF, see S3 Fig).

Similar LME models were fitted for each single node of each tract profile and they confirmed this pattern of results (Figs 5 and 6); there were no observable changes in white matter properties over the two-week training period in either groups. This was confirmed even in the central parts of the tracts, which are less likely to be affected by volume conduction artifacts.

To estimate the evidence supporting the null hypothesis (H0: no plasticity), additional Bayesian analyses were run on each tract to compare the dMRI training effect (post-pre mean profile difference) between the two groups. Bayes factors supported small-to-moderate evidence for the null effect in the majority of the tracts, including all tracts that are part of the reading circuitry (FA: BFs < 1 in 16 of the 18 tracts; MD: BFs < 1 in 12 of the 18 tracts; Fig 7).

**Linking training effects between behavioral and dMRI measures.** Pearson correlations were calculated to check whether individual changes in alphabet knowledge and decoding could be mapped onto structural changes of two major reading white matter tracts: the left AF and left ILF. No significant effect was present after multiple comparisons correction (see Table 5). However, BFs signaled a moderate support for the presence of a link between Alphabet

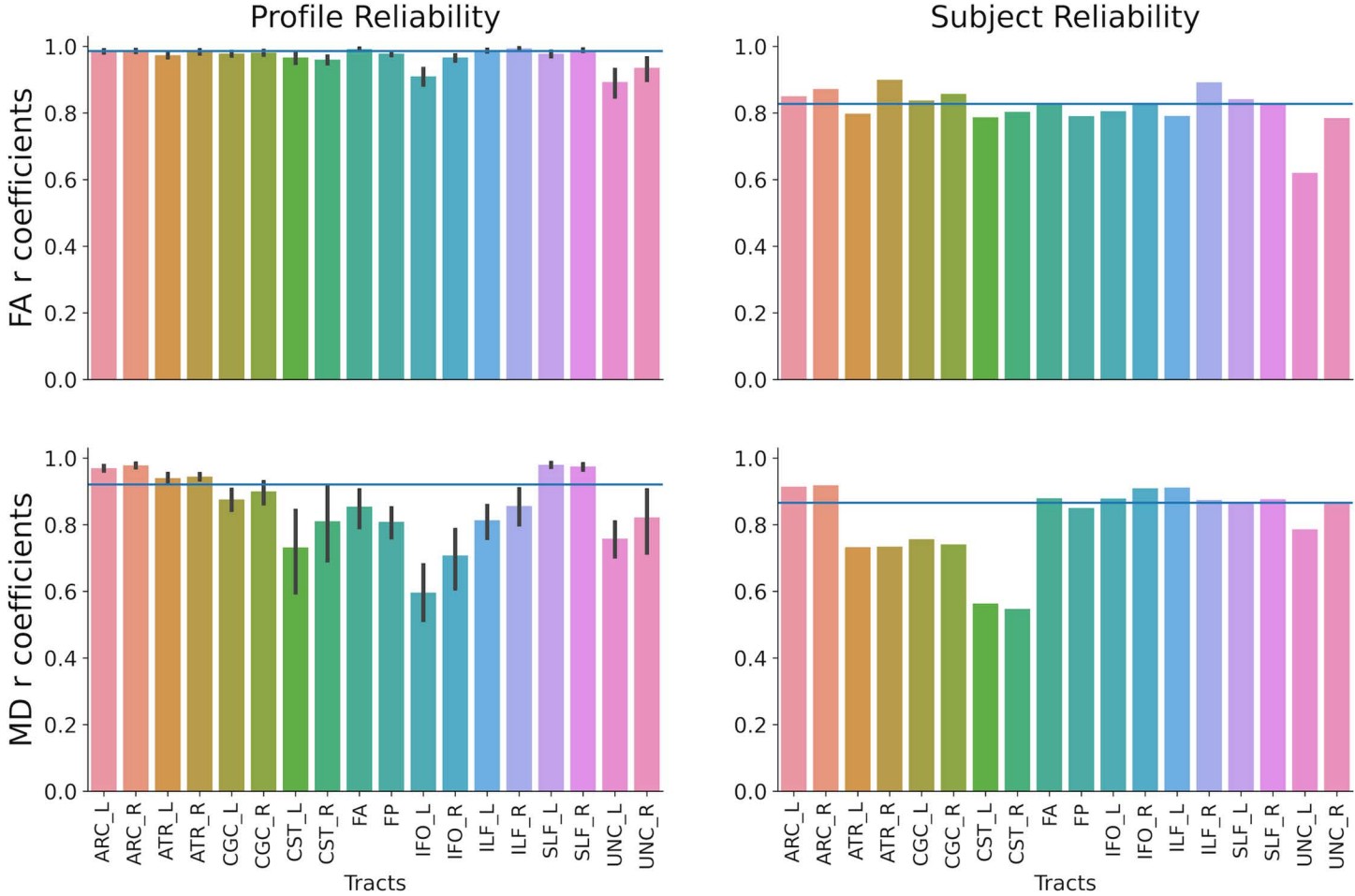

**Fig 3. Scan Rescan reliability.** The two columns show the profile and subject reliability estimates of each white matter tract examined in the study. The two rows show reliability estimates for FA and MD, respectively. ARC: Arcuate Fasciculus; ATR: Anterior Thalamic Radiation; CGC: Cingulum Cingulate; CST: Corticospinal Tract; FA: Anterior Forceps; FP: Posterior Forceps; IFO: Inferior Longitudinal Fasciculus; ILF: Inferior Longitudinal Fasciculus; SLF: Superior Longitudinal Fasciculus; UNC: Uncinate.

knowledge and FA of the left ILF (BF > 3), with smaller improvements of orthographic knowledge corresponding to greater FA values. This counterintuitive trend might be due to the fact that kids showing small changes in Alphabet knowledge (and greater FA changes) were also those having high levels of Alphabet knowledge at the pre-training session, leaving them with a small margin for measurable improvement. This ceiling effect in the Alphabet knowledge scale might have hindered our ability to accurately measure the behavioral improvement of those kids that showed greater FA changes.

## Discussion

This randomized controlled trial examined short-term effects of a Letter and a Language training program on preschoolers' reading performance and brain structure. The findings suggest that a two-week Letter training program causes improvements in preschoolers' letter knowledge and decoding skills. However, this behavioral effect was not accompanied by short-term changes in the diffusion properties (i.e., FA and MD) of white matter pathways, within or outside the reading circuitry. The presence of quick behavioral changes as a result of

# Left Arcuate - dMRI results

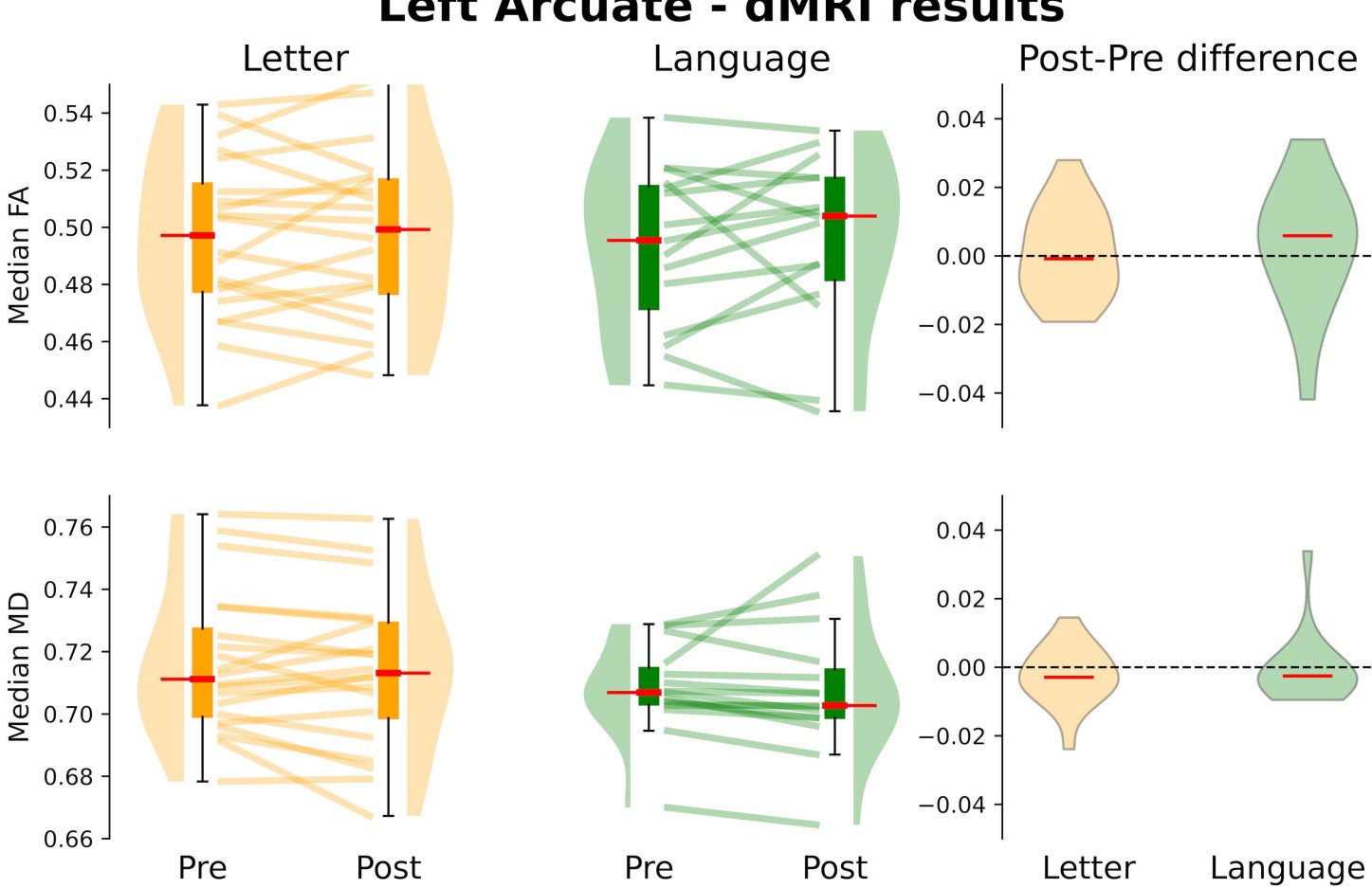

**Fig 4. Training-related dMRI changes of the left arcuate.** First two columns: structural changes of the left arcuate are shown for the Letter and Language Training groups and each experimental session. The third column shows the distribution of FA and MD changes (i.e., difference between the individual profiles observed in the post and pre-training sessions) for each group.

Letter training confirms previous findings on the effectiveness of short-term reading instruction, which has been observed in children with and without reading disorders [57–60].

Our dMRI findings further complement the existing literature on short-term reading-relating brain plasticity by showing that reading performance improvements are not always accompanied by changes in diffusion properties of white matter pathways [4,5,16,24,25]. The high reliability estimates for both FA and MD scores across sessions ensure that this null effect could not be accounted for by low dMRI data quality. Bayesian analyses provided support for the null hypothesis (no change in white matter diffusion) for all major white matter tracts of the reading network. In addition, correlation analyses did not show a clear coupling between preschoolers' individual behavioral improvements and variations in structural properties of reading white matter pathways.

One aspect that can account for the lack of structural changes is the type and intensity of the reading program. Previous studies have mainly focused on effects of reading intervention in children diagnosed with dyslexia, which can have an intense and profound impact on struggling readers' cognitive and social lives. In the current study, our reading training proposed preschool/kindergarten activities that are usually carried out in a classroom setting. Since

**FA Tract profile comparisons**

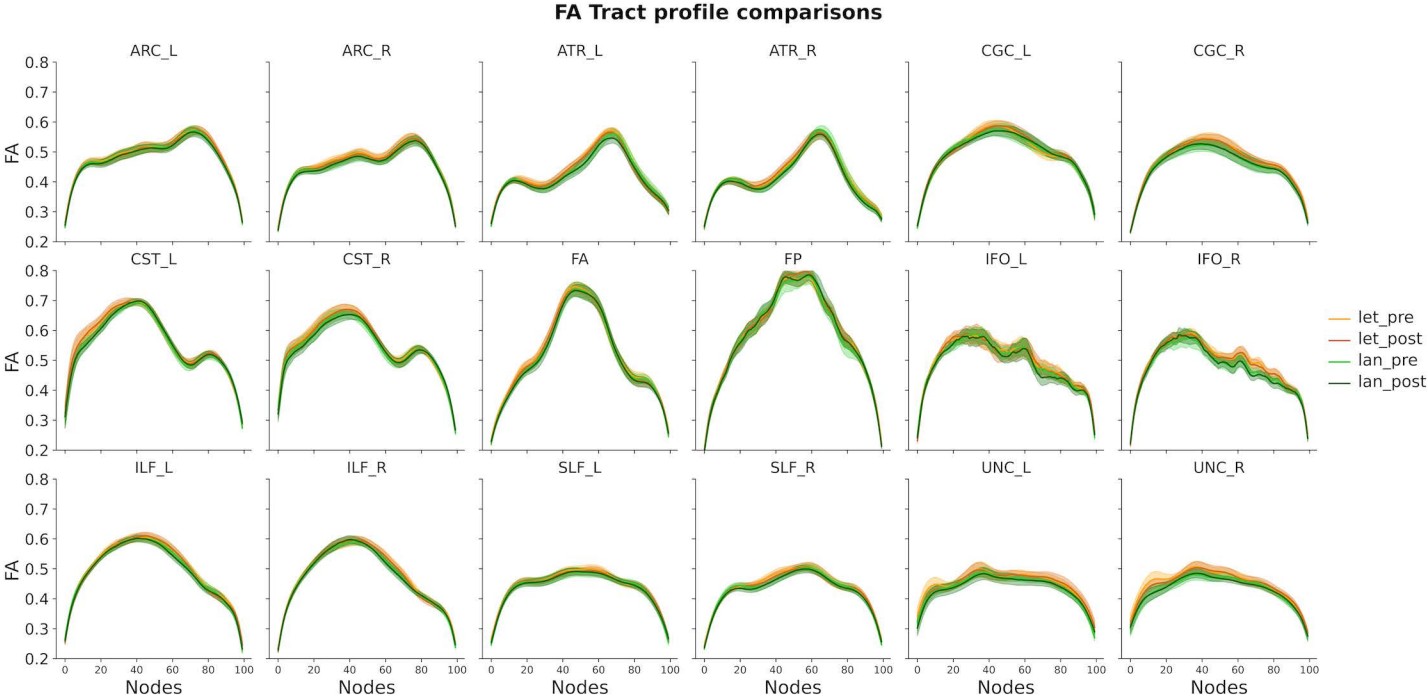

**Fig 5. FA tract profile for each experimental group and training session.** The plots show FA values estimated based on the beta coefficients extracted from node-by-node LME models. Shaded areas represent +/- 2 SE. ARC: Arcuate Fasciculus; ATR: Anterior Thalamic Radiation; CGC: Cingulum Cingulate; CST: Corticospinal Tract; FA: Anterior Forceps; FP: Posterior Forceps; IFO: Inferior Longitudinal Fasciculus; ILF: Inferior Longitudinal Fasciculus; SLF: Superior Longitudinal Fasciculus; UNC: Uncinate.

these training programs are similar to common preschool and kindergarten classrooms, they might not represent a dramatic enough environmental change to cause large-scale remodeling of the white matter.

Another variable that can account for our results is the cognitive profile of the trainees. This is the first randomized controlled trial on short-term reading training with typically developing preschoolers. Previous experimental evidence collected so far (Table 1) refers to the effects of short-term remediation programs on children with reading disorders or at-risk of reading disorders. Hence, the large effects that have been reported so far might reflect the dramatic change of entering an intensive intervention environment after struggling in school for years. It is also possible that the effects previously reported mainly reflect compensatory mechanisms put in place by children with (or at risk for) dyslexia. Previous findings have shown that dyslexics' white matter pathways have different microstructural properties from those of controls even before reading instruction has begun [14,61,62]. Hence, the effect of intervention on dyslexics' brain structure might be driven by an adaptive response of an already divergent system. This experience is quite different than typically developing children beginning formal reading instruction [3–5,23].

Another possible explanation to consider is the type of diffusion properties examined here. Recent dMRI findings on the short-term effects of reading intervention programs in preschoolers reported structural changes only in myelin water fraction, but not in FA and MD scores [3,25]. This might suggest that MRI measures more specifically related to myelination would better reflect reading-related short-term plasticity around 5 years of age. However, within 7 and 12 years of age an opposite pattern of results have been reported, with MD and FA providing evidence for rapid structural plasticity while no training-dependent changes

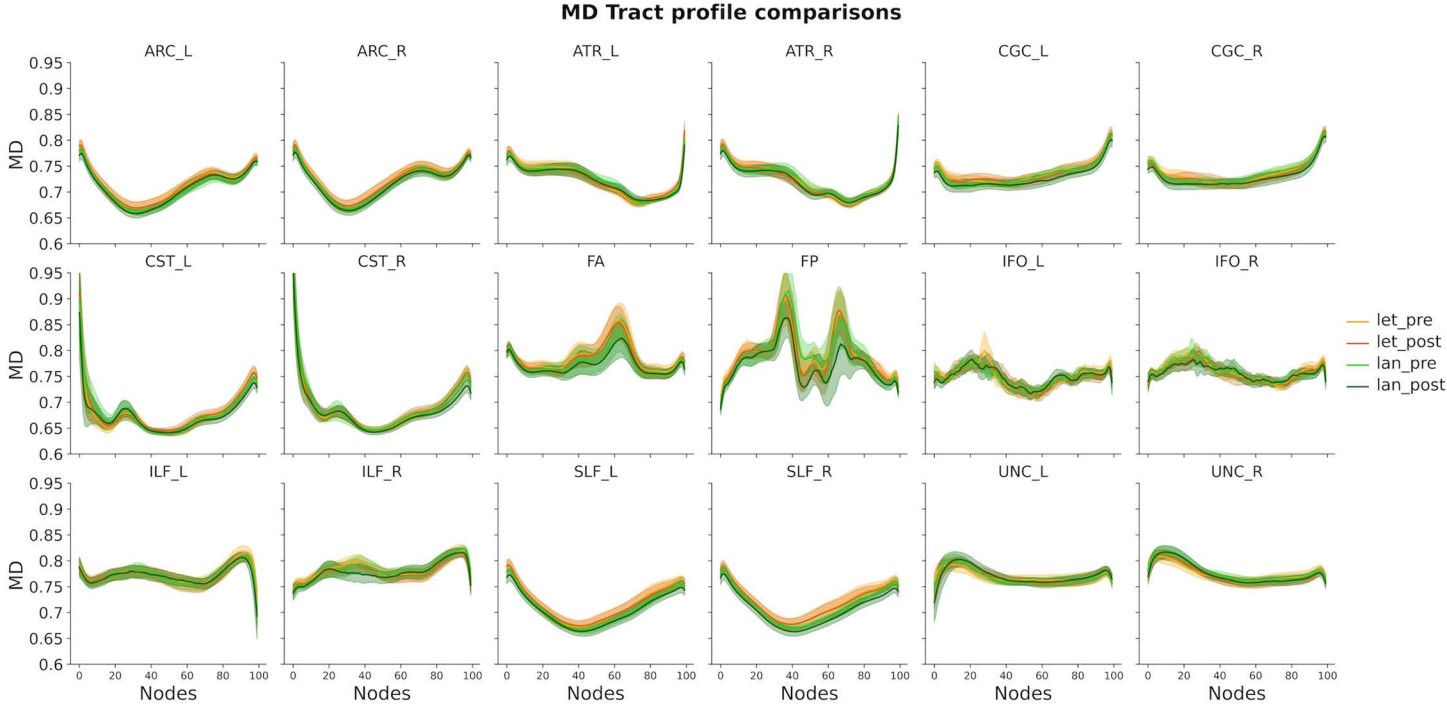

**Fig 6. MD tract profile for each experimental group and training session.** The plots show MD values estimated based on the beta coefficients extracted from node-by-node LME models. Shaded areas represent +/- 2 SE. ARC: Arcuate Fasciculus; ATR: Anterior Thalamic Radiation; CGC: Cingulum Cingulate; CST: Corticospinal Tract; FA: Anterior Forceps; FP: Posterior Forceps; IFO: Inferior Longitudinal Fasciculus; ILF: Inferior Longitudinal Fasciculus; SLF: Superior Longitudinal Fasciculus; UNC: Uncinate.

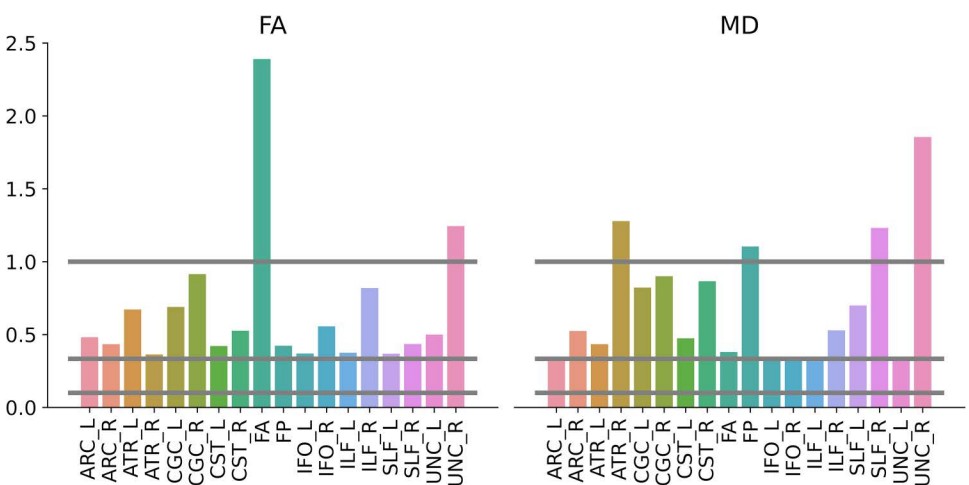

**Fig 7. Bayes factors relative to the group comparison of the dMRI training effect for each tract.** ARC: Arcuate Fasciculus; ATR: Anterior Thalamic Radiation; CGC: Cingulum Cingulate; CST: Corticospinal Tract; FA: Anterior Forceps; FP: Posterior Forceps; IFO: Inferior Longitudinal Fasciculus; ILF: Inferior Longitudinal Fasciculus; SLF: Superior Longitudinal Fasciculus; UNC: Uncinate.

**Table 5. Pearson correlations between post-pre differences in reading performance and structural properties.**

| | FA | | | MD | | |
|---|---|---|---|---|---|---|
| **Alphabet knowledge** | **r** | **p corr** | **BF** | **r** | **p corr** | **BF** |
| Left AF | -0.38 | 0.42 | 2.40 | 0.12 | 0.99 | 0.26 |
| Left ILF | -0.41 | 0.24 | 3.72 | 0.36 | 0.50 | 1.84 |
| **Decoding** | **r** | **p corr** | **BF** | **r** | **p corr** | **BF** |
| Left AF | -0.11 | 0.99 | 0.25 | -0.14 | 0.99 | 0.29 |
| Left ILF | -0.05 | 0.99 | 0.22 | -0.07 | 0.99 | 0.23 |

were reported for more myelin-specific correlates, such as axonal water fraction and R1 [4,23]. These results are still compatible with the idea that short-term plasticity due to reading training might affect different structural properties of white matter depending on the developmental time window (e.g., there might be a higher degree of plasticity for myelin-specific indexes in the early stages of life). Additional research is needed in order to clarify which type of diffusion properties can be shaped by experience as a function of age (e.g., neural or non-neuronal plasticity, intra or extra axonal plasticity, [3,23,63]).

Finally, another potential explanation for our dMRI findings regards the presence of a possible time shift between the training effects on behavior and brain structure, with white matter changes happening over a larger temporal scale compared to behavioral changes. For instance, our findings are still compatible with the idea that at this early age the amount of training received is not sufficient to shape what will become the reading circuit later on. Although some studies have shown no time lag between behavioral and structural changes in response to a short reading intervention program [3–5,16], the exact time course of reading-related neuroplasticity is still understudied and needs further investigation.

Overall, this heterogeneous picture of findings on short-term reading-related structural neuroplasticity highlights the need to better define the conditions under which white matter can be shaped by experience. Several experiential and developmental factors might modulate the degree of white matter plasticity exhibited in response to reading training or intervention. Research evidence coming from other cognitive domains might give us some insights on the critical constraining variables to be considered. For instance, studies testing for the presence of a sensitive period of sensory and motor white matter circuits suggest that the time onset of the environmental exposure is a key factor to establish whether white matter structure is stable or plastic [64,65]. Other factors that have been suggested to modulate the balance between structural plasticity and stability are the type and the duration of experiential exposure [66–70], the individual cognitive health and lifestyle risk factors [68,71].

In conclusion, this randomized controlled trial highlights that a two-week literacy training can cause fast behavioral changes in preschoolers' reading performance without being accompanied by fast FA and MD changes of the reading circuitry. These findings highlight that rapid diffusion properties variations are not always observed in response to short-term reading training and point to the need of specifying the conditions under which white matter structure is plastic versus stable.

## Supporting information

**S1 Fig. Subject reliability of FA for each experimental group.**
(TIFF)

**S2 Fig. Subject reliability of MD for each experimental group.**
(TIFF)

**S3 Fig. Training-related dMRI changes of the left inferior fasciculus.** First two columns: structural changes of the left ILF are shown for the Letter and Language Training groups and each experimental session. The third column shows the distribution of FA and MD changes (i.e., difference between the individual profiles observed in the post and pre-training sessions) for each group.
(TIFF)

## Author contributions

**Conceptualization:** Sendy Caffarra, Jason D. Yeatman.

**Data curation:** Sendy Caffarra, Iliana I. Karipidis, Emily Kubota, Megumi Takada.

**Formal analysis:** Sendy Caffarra, Iliana I. Karipidis, Emily Kubota, Megumi Takada.

**Funding acquisition:** Jason D. Yeatman.

**Methodology:** John Kruper, Adam Richie-Halford, Ariel Rokem.

**Project administration:** Sendy Caffarra.

**Resources:** Jason D. Yeatman.

**Software:** John Kruper, Adam Richie-Halford, Ariel Rokem.

**Supervision:** Ariel Rokem, Jason D. Yeatman.

**Writing – original draft:** Sendy Caffarra.

**Writing – review & editing:** Iliana I. Karipidis, John Kruper, Emily Kubota, Adam Richie-Halford, Ariel Rokem, Jason D. Yeatman.

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
