## [Decision Letter · Decision Letter 0]

27 Sep 2024

PONE-D-24-32615Assessing white matter plasticity in a randomized controlled trial of early literacy training in preschoolersPLOS ONE

Dear Dr. Caffarra,

Thank you for submitting your manuscript to PLOS ONE. After careful consideration, we feel that it has merit but does not fully meet PLOS ONE’s publication criteria as it currently stands. Therefore, we invite you to submit a revised version of the manuscript that addresses the points raised during the review process.

We look forward to receiving your revised manuscript.

Kind regards,

Signe Bray

Academic Editor

PLOS ONE

Journal Requirements:

“SC conducted this research as part of the projects FAR Mission Oriented 2022 and PRIN PNRR 2022 P2022SMEJW, which were funded by the European Union – Next Generation EU. IK was supported by the Stanford Maternal and Child Health Research Institute award. AR and JK were funded by NSF grant 1934292 and NIH grants RF1 MH121868 and R01EB027585. JK was additionally supported through the NSF Graduate Research Fellowship DGE-2140004, NIH grants R21HD092771 and R01HD095861.”

5.  We note you have included a table to which you do not refer in the text of your manuscript. Please ensure that you refer to Table 2 in your text; if accepted, production will need this reference to link the reader to the Table.

**Additional Editor Comments:**

This work has now been reviewed by two experts in the field. While both reviewers felt that this work addresses an important topic, they have several suggestions to improve context, clarity and interpretation.

Reviewers' comments:

Reviewer's Responses to Questions

**Comments to the Author**

1. Is the manuscript technically sound, and do the data support the conclusions?

Reviewer #1: Yes

Reviewer #2: Yes

2. Has the statistical analysis been performed appropriately and rigorously? 

Reviewer #1: Yes

Reviewer #2: Yes

3. Have the authors made all data underlying the findings in their manuscript fully available?

Reviewer #1: No

Reviewer #2: No

4. Is the manuscript presented in an intelligible fashion and written in standard English?

Reviewer #1: Yes

Reviewer #2: Yes

5. Review Comments to the Author

Reviewer #1: Reviewer Comments

The manuscript “Assessing white matter plasticity in a randomized controlled trial of early literacy training in preschoolers” describes a randomly controlled trial which showed literacy skill improvement in preschool children who participated in letter-focused literacy training, but not those who participated in oral language training. This study supports the efficacy of short-term literacy training in typically developing children. No corresponding changes in diffusion-tensor imaging metrics of white matter microstructure were found. This study fills a gap in the literature, which has mainly focused on training and intervention programs for children with or at-risk for reading disabilities. This manuscript would benefit from further development of the methods to provide clarity. My specific comments are detailed below:

Major Comments:

1. Were the same versions/forms of the assessments used for behavioral assessment before and after the training period? Were any measures taken to account for practice effects on the assessments?

2. Please provide a detailed description of the statistical analysis (LME models for both behavioral and dMRI measures) in the Methods section. How many total models were tested? Were multiple comparisons controlled? Were any model fitting parameters computed to verify whether including random slopes improved model fit (was there substantial random effects variance in the models)? Please also justify why random slopes were included in the behavioral models but not the dMRI models. Sharing code would be helpful here.

3. Were any analyses conducted to test whether the groups were well-matched on language and reading abilities at baseline? Were the numbers of males and females matched across groups (This is relevant given that girls tend to perform better than boys on language measures at this young age)? It would be helpful to include a table indicating the demographic and behavioral characteristics of the groups prior to training, including whether the time between MRI sessions and training (pre & post) were similar across groups.

4. The distinction between the “profile reliability” and “subject reliability” is not clear. Specifically, please clarify what value(s) is used to calculate profile reliability and what is meant by “tract profile”. Is an average taken across nodes to calculate this reliability, or does the profile reliability describe reliability within nodes across participants?

5. Please clarify in the Results whether there were any significant main effects of time (i.e., improvement regardless of training group).

6. In the methods section, it is stated that DTI metrics were weighted by streamlines’ distance from the tract core. Based on examination of the tract profiles, FA values at the tails are much lower than along the main body of the tract; some tracts (e.g. ILF) show extremely high values for MD at the tails. Was any weighting applied to account for averaging across nodes along the tract to account for distance from the central node and the extreme FA/MD values observed at the tails for many tracts?

7. For the follow-up analysis “Linking training effects between behavioral and dMRI measures”, please clarify why this analysis focused only on alphabet knowledge (not other behavioral measures) and only on the left AF and ILF. Please also include a description of this analysis in the methods section.

8. In the discussion, please consider whether the lack of dMRI affects in this typically developing preschool sample could be because the children’s brains are already well-wired for reading before formal reading instruction begins. This would be consistent with evidence that white matter differences in children with and without (risk for) dyslexia are observed prior to formal reading instruction, and even in infancy. Thus, dramatic and rapid reading-instruction-related white matter plasticity may not be expected in this sample, in contrast to children at risk for dyslexia who may require white matter adaptation to support development of an adequate reading network.

Minor comments:

1. In the introduction, only the AF and ILF are introduced as important tracts for reading, however, several other tracts have been linked to reading and this study includes 18 white matter pathways within and outside the reading network. Additional tracts including the SLF and IFOF should be mentioned.

2. Table 1 provides a helpful summary of the prior literature, it would be useful to index the papers by author & year in addition to the reference numbers.

3. In the “behavioral data acquisition” section (page 11), please include the names of the full assessments in the main text along with their citations.

4. Please provide a reference for the OASIS target template (page 12).

5. In Figure 1, it appears that pre-training and post training sessions occurred immediately at the onset and completion of the 2-week training. For clarity, please show in the figure that these sessions occurred over a range of time prior to and after training.

Reviewer #2: This randomized controlled study investigated the effects of two training programs—language and literacy—on reading-related skills in English-speaking preschoolers. The authors evaluated behavioral outcomes and changes in white matter integrity using diffusion MRI (dMRI) metrics, specifically fractional anisotropy (FA) and mean diffusivity (MD) across 18 white matter tracts, including the left arcuate fasciculus and left inferior longitudinal fasciculus. Forty-eight preschoolers (mean age 5 years) were randomly assigned to either a Letter Program or a Language Program, both conducted in a summer camp format over two weeks, totaling 30 hours of training. Behavioral and MRI data were collected before and after the intervention.

Linear mixed-effects models revealed a significant interaction between Training Type and Time for alphabet knowledge and decoding skills. Post-hoc analyses indicated that children in the Letter Program showed significant improvements in both alphabet knowledge and decoding skills, whereas no such effects were observed in the Language Program. In contrast, dMRI analyses showed no significant interaction effects or main effects of Time or Training Type, indicating no structural changes in the white matter. Bayesian analyses further supported the null hypothesis, with Bayes factors providing small-to-moderate evidence against structural changes in most of the white matter tracts examined.

Given the limited research on structural brain changes in typically developing preschoolers, this study contributes to the literature by showing that a relatively short, intensive intervention (30 hours over two weeks) can lead to behavioral improvements without corresponding structural changes in the brain. These findings highlight the need for further research to explore how the duration and intensity of interventions impact both behavioral outcomes and brain structure over time.**

Suggestions for Strengthening the Manuscript:

Introduction:

The study's hypotheses and predictions are not clearly articulated. In Table 1, only Study [3] reported structural changes in preschoolers following a reading intervention, while another study in the same age group did not. Other studies listed in Table 1 reported structural changes only with higher dosage interventions (>100 hours) and in school-aged children. It would be helpful to clarify whether structural changes were anticipated in this study's preschool cohort, particularly in relation to findings from [21] and [20]. Although the two-week, 30-hour intervention is intensive for preschoolers, the relationship between dosage, intervention length, and observed effects should be more explicitly discussed. Clarifying these concepts, as well as outlining the study's hypotheses and predictions, would enhance the theoretical framework of the manuscript.

Participants:

To better contextualize the sample, the manuscript should provide demographic details, including the mean age (with standard deviations) and gender distribution for both groups. Although the inclusion criteria are well defined, it is unclear whether the groups were balanced on behavioral measures before the intervention. Presenting descriptive statistics for pre-intervention behavioral measures would clarify this. Additionally, the authors should explicitly confirm whether raw scores were used for analyses. It would be beneficial to include both pre- and post-intervention descriptive statistics for all behavioral measures, even for those that did not show significant changes. Including effect sizes is critical, as this would help address whether the absence of structural changes is due to the short intervention period or because the treatment effect was too small to induce measurable structural changes.

Program Fidelity:

The manuscript would benefit from more details on how program fidelity was assessed to ensure that the training was consistently delivered across participants.

Measure and Figure:

It is unclear which task the authors refer to as “decoding skills” in the results. Is this referring to the phonological awareness literacy screening task? Regarding Figure 2, distinguishing between uppercase and lowercase letters may not add value if the analysis is based on their average score. Since significant results were found for alphabet knowledge (using the average score of uppercase and lowercase letters) and decoding skills, it would be more informative to present these two variables in the figure.

Results and Interpretation (Linking Training Effects to dMRI Measures):

The factor controlled for in the Bonferroni correction in Table 3 is not clear. Providing a more detailed explanation of the correction method would enhance clarity. Additionally, given that the correlation coefficients (-0.38, -0.41, 0.36) are moderate, the non-significant results could be attributed to the limited sample size. Since correlation coefficients can be interpreted as effect sizes, these moderate values suggest the potential need for further exploration. The authors may consider running Bayesian analyses as a validation step to provide additional insight into these findings.

6. PLOS authors have the option to publish the peer review history of their article (what does this mean?). If published, this will include your full peer review and any attached files.

Reviewer #1: No

Reviewer #2: **Yes: **Silvia Clement-Lam

---

## [Author Response · Author response to Decision Letter 1]

11 Nov 2024

Journal Requirements:

We confirm that our manuscript is formatted in accordance with these guidelines.

“SC conducted this research as part of the projects FAR Mission Oriented 2022 and PRIN PNRR 2022 P2022SMEJW, which were funded by the European Union – Next Generation EU. IK was supported by the Stanford Maternal and Child Health Research Institute award. AR and JK were funded by NSF grant 1934292 and NIH grants RF1 MH121868 and R01EB027585. JK was additionally supported through the NSF Graduate Research Fellowship DGE-2140004, NIH grants R21HD092771 and R01HD095861.”

We have specified in the Cover letter that “The funders had no role in study design, data collection and analysis, decision to publish, or preparation of the manuscript.”

3. We note that you have indicated that there are restrictions to data sharing for this study. For studies involving human research participant data or other sensitive data, we encourage authors to share de-identified or anonymized data. However, when data cannot be publicly shared for ethical reasons, we allow authors to make their data sets available upon request.

The de-identified dataset is now available here: doi:10.18112/openneuro.ds005572.v1.0.0

We updated our Data Availability statement in the submission form accordingly.

4. We note you have included a table to which you do not refer in the text of your manuscript. Please ensure that you refer to Table 2 in your text; if accepted, production will need this reference to link the reader to the Table.

We added a reference to Table 2 (now Table 3) in the Procedure section (page 9).

Reviewers' comments:

Reviewer #1:

Major Comments:

1. Were the same versions/forms of the assessments used for behavioral assessment before and after the training period? Were any measures taken to account for practice effects on the assessments?

The same behavioral assessment was carried out for both groups pre- and post training. However, our main findings hold even if practice effects were present. The experimental design of the randomized controlled trial enables us to separate practice effects (indexed by a main effect of Time) from training-specific effects (signaled by a significant interaction Training Type x Time). Before training no behavioral difference was observed between groups (this information is now added in Table 2). Any potential practice effect should have been present in a similar way for both training groups (i.e., Letter and Language). The significant interaction between Training Type and Time demonstrates that the behavioral improvements are specifically caused by the Letter training program and this effect is present over and beyond the effect of repeated practice.

2. Please provide a detailed description of the statistical analysis (LME models for both behavioral and dMRI measures) in the Methods section. How many total models were tested? Were multiple comparisons controlled? Were any model fitting parameters computed to verify whether including random slopes improved model fit (was there substantial random effects variance in the models)? Please also justify why random slopes were included in the behavioral models but not the dMRI models. Sharing code would be helpful here.

The description of the statistical analysis was moved from the Results to the respective Methods sections (page 14 for the behavioral analyses and pages 17-18 for the dMRI analyses). No multiple comparisons corrections were applied across models. However, even after applying the most conservative multiple comparison correction (Bonferroni corrected p value threshold for significance: 0.004) the significant behavioral improvement observed in alphabet knowledge and decoding skills for the Letter group still hold (alphabet knowledge: β = 2.770, SE = 0.784, t = 3.532, p < 0.001, Cohen’s d = 0.406; decoding: and decoding ability (β = 1.689, SE = 0.496, t = 3.403, p = 0.001, Cohen’s d = 0.855). Descriptions of model comparisons based on Bayesian Information Criteria were included on pages 14 and 17. Random slopes improved model fit only in the case of behavioral analyses and they were included in the LME models for this reason. The code used to analyze this data is now available here https://github.com/SendyCaffarra/PREK-analysis.git

3. Were any analyses conducted to test whether the groups were well-matched on language and reading abilities at baseline? Were the numbers of males and females matched across groups (This is relevant given that girls tend to perform better than boys on language measures at this young age)? It would be helpful to include a table indicating the demographic and behavioral characteristics of the groups prior to training, including whether the time between MRI sessions and training (pre & post) were similar across groups.

Thank you for this suggestion. Table 2 was added to summarize the demographic characteristics (age, gender and MRI-training time lag) and the behavioral scores (reading and verbal skills) of the two groups before training (pages 9 and 10). Statistical comparisons are also reported, showing no differences between the Letter and the Language groups before training.

4. The distinction between the “profile reliability” and “subject reliability” is not clear. Specifically, please clarify what value(s) is used to calculate profile reliability and what is meant by “tract profile”. Is an average taken across nodes to calculate this reliability, or does the profile reliability describe reliability within nodes across participants?

Tract profile represents all FA (or MD) values calculated for each node along a tract of a single participant. We added this definition to the Materials and Methods section (page 17). Profile reliability is first calculated at a subject-level as the correlation between the tract profiles of the pre and post-training sessions. These correlation coefficients are finally averaged across participants to obtain the final profile reliability score.

To calculate the subject reliability, we first obtained the median value across the 100 nodes of a single tract and participant (individual tract value). We then calculated the Pearson correlation between the individual tract values of the pre-training session and the individual tract values of the post-training session. We updated the description on profile reliability and subject reliability on page 17.

5. Please clarify in the Results whether there were any significant main effects of time (i.e., improvement regardless of training group).

We added statistical details of the main effect of Time and Training Type on pages 19. There was no effect of Training Type indicating no overall between-group differences. A main effect of Time was significant indicating a general improvement of alphabet knowledge and decoding scores after training. This main effect could partially reflect practice effects. The significant interaction Training Type x Time highlighted that there was also a training-specific behavioral improvement.

6. In the methods section, it is stated that DTI metrics were weighted by streamlines’ distance from the tract core. Based on examination of the tract profiles, FA values at the tails are much lower than along the main body of the tract; some tracts (e.g. ILF) show extremely high values for MD at the tails. Was any weighting applied to account for averaging across nodes along the tract to account for distance from the central node and the extreme FA/MD values observed at the tails for many tracts?

No weighting was applied while averaging the FA (or MD) values across nodes. However, our dMRI findings were replicated even when LME models tested the effects of our experimental conditions on each single node of the tract. These models independently examined parts of the tracts that were less likely to be affected by volume conduction artifacts and led to the same results. We added an additional elaboration on this point on page 22.

7. For the follow-up analysis “Linking training effects between behavioral and dMRI measures”, please clarify why this analysis focused only on alphabet knowledge (not other behavioral measures) and only on the left AF and ILF. Please also include a description of this analysis in the methods section.

Thank you for this suggestion. We clarified that these follow up analyses focused on two major reading white matter pathways and included all behavioral measures that showed a significant training effect. A description of the rationale of these analyses was included in the Methods section (page 18). We added the correlations relative to the decoding scores to give a more general overview of all training-specific behavioral effects and their potential coupling with white matter (page 24).

8. In the discussion, please consider whether the lack of dMRI effects in this typically developing preschool sample could be because the children’s brains are already well-wired for reading before formal reading instruction begins. This would be consistent with evidence that white matter differences in children with and without (risk for) dyslexia are observed prior to formal reading instruction, and even in infancy. Thus, dramatic and rapid reading-instruction-related white matter plasticity may not be expected in this sample, in contrast to children at risk for dyslexia who may require white matter adaptation to support development of an adequate reading network.

Thank you for this suggestion. We added this point to the Discussion on page 26.

Minor comments:

1. In the introduction, only the AF and ILF are introduced as important tracts for reading, however, several other tracts have been linked to reading and this study includes 18 white matter pathways within and outside the reading network. Additional tracts including the SLF and IFOF should be mentioned.

We updated our Introduction mentioning a larger list of white matter tracts that have been related to reading abilities, including SLF and IFOF (pages 3 and 4).

2. Table 1 provides a helpful summary of the prior literature, it would be useful to index the papers by author & year in addition to the reference numbers.

We updated Table 1 following this suggestion.

3. In the “behavioral data acquisition” section (page 11), please include the names of the full assessments in the main text along with their citations.

We updated the “behavioral data acquisition” section following this suggestion (page 13). We also added a more extensive description of each behavioral task.

4. Please provide a reference for the OASIS target template (page 12).

We added the following reference on page 15: D.S. Marcus, T.H. Wang, J. Parker, J.G. Csernansky, J.C. Morris, R.L. Buckner. Open Access Series of Imaging Studies (OASIS): cross-sectional MRI data in young, middle aged, nondemented, and demented older adults. J. Cogn. Neurosci., 19 (9) (2007), pp. 1498-1507

5. In Figure 1, it appears that pre-training and post training sessions occurred immediately at the onset and completion of the 2-week training. For clarity, please show in the figure that these sessions occurred over a range of time prior to and after training.

Figure 1 was updated to show the time range of pre- and post- training data collection.

Reviewer #2:

1. Introduction:

The study's hypotheses and predictions are not clearly articulated. In Table 1, only Study [3] reported structural changes in preschoolers following a reading intervention, while another study in the same age group did not. Other studies listed in Table 1 reported structural changes only with higher dosage interventions (>100 hours) and in school-aged children. It would be helpful to clarify whether structural changes were anticipated in this study's preschool cohort, particularly in relation to findings from [21] and [20]. Although the two-week, 30-hour intervention is intensive for preschoolers, the relationship between dosage, intervention length, and observed effects should be more explicitly discussed. Clarifying these concepts, as well as outlining the study's hypotheses and predictions, would enhance the theoretical framework of the manuscript.

Thank you for this suggestion. We updated the Introduction to clarify that, based on the existing literature, it is still unclear whether high dosage intervention is essential to see reading-related structural neuroplasticity in children. We highlighted that some studies adopting 100-hour intervention reported structural changes even before the end of the program. We updated the notes of Table 1 to mark those studies where structural changes were reported not only at the end but already within the first 50 hours of the intervention (duration range: 18-46 hours). We updated the main text of the Introduction (page 7) to highlight that the minimal duration of intervention required to see white matter changes is still not fully clear. We also updated the final part of the Introduction to state clearly our hypotheses based on these considerations (page 8).

2. Participants:

To better contextualize the sample, the manuscript should provide demographic details, including the mean age (with standard deviations) and gender distribution for both groups. Although the inclusion criteria are well defined, it is unclear whether the groups were balanced on behavioral measures before the intervention. Presenting descriptive statistics for pre-intervention behavioral measures would clarify this. Additionally, the authors should explicitly confirm whether raw scores were used for analyses. It would be beneficial to include both pre- and post-intervention descriptive statistics for all behavioral measures, even for those that did not show significant changes. Including effect sizes is critical, as this would help address whether the absence of structural changes is due to the short intervention period or because the treatment effect was too small to induce measurable structural changes.

Thank you for this suggestion. Table 2 now provides mean age and gender distribution of each group. Descriptive statistics and between-group comparisons are also reported in the same Table. No behavioral differences emerged across groups prior to training. We specified that raw scores of each behavioral test were used in the statistical analyses on page 14. The “Behavioral results” section now reports null statistical details for each behavioral test (Table 4). Effect sizes are now reported in the Results sections (pages 18 and 19) indicating medium-to-large effect sizes at the behavioral level.

3. Program Fidelity:

The manuscript would benefit from more details on how program fidelity was assessed to ensure that the training was consistently delivered across participants.

Additional details about program implementation and fidelity are now provided in the procedure section (page 11): “To maximize consistency of the training delivery over time and across participants the same group of teachers administered both Letter and Language activities. In addition, two pilot camps were run before the experiment to reach high levels of coherence among teachers’ styles. During these pilot camps of four days the three teachers could practice and coordinate to minimize differences in the way each activity was delivered.”

4. Measure and Figure:

It is unclear which task the authors refer to as “decoding skills” in the results. Is this referring to

---

## [Decision Letter · Decision Letter 1]

10 Dec 2024

PONE-D-24-32615R1Assessing white matter plasticity in a randomized controlled trial of early literacy training in preschoolersPLOS ONE

Dear Dr. Caffarra,

Thank you for submitting your manuscript to PLOS ONE. After careful consideration, we feel that it has merit but does not fully meet PLOS ONE’s publication criteria as it currently stands. Therefore, we invite you to submit a revised version of the manuscript that addresses the points raised during the review process.

 Both reviewers feel that the manuscript is substantially improved and have requested only very minor changes in a revision.

We look forward to receiving your revised manuscript.

Kind regards,

Signe Bray

Academic Editor

PLOS ONE

Journal Requirements:

Reviewers' comments:

Reviewer's Responses to Questions

**Comments to the Author**

1. If the authors have adequately addressed your comments raised in a previous round of review and you feel that this manuscript is now acceptable for publication, you may indicate that here to bypass the “Comments to the Author” section, enter your conflict of interest statement in the “Confidential to Editor” section, and submit your "Accept" recommendation.

Reviewer #1: (No Response)

Reviewer #2: All comments have been addressed

2. Is the manuscript technically sound, and do the data support the conclusions?

Reviewer #1: Yes

Reviewer #2: Yes

3. Has the statistical analysis been performed appropriately and rigorously? 

Reviewer #1: Yes

Reviewer #2: Yes

4. Have the authors made all data underlying the findings in their manuscript fully available?

Reviewer #1: No

Reviewer #2: Yes

5. Is the manuscript presented in an intelligible fashion and written in standard English?

Reviewer #1: Yes

Reviewer #2: Yes

6. Review Comments to the Author

Reviewer #1: The reviewers have carefully addressed all of my initial queries. I have several small suggestions pertaining to the revised version of the manuscript:

Data Availability: I attempted to access the data at doi: 10.18112/openneuro.ds005572.v1.0.0 and received the error “403: You do not have access to this page, you may need to sign in.” I was still unable to access the data after signing in with my ORCID.

Table 2.

- Typo: Post MRI-camp time different (days)

- Please indicate whether Raw or Standard/Scaled scores are reported (also for the PPVT inclusion criteria, p.8 l. 147).

Methods

- Please cite software and packages used for statistical analysis.

- Thank you for providing additional clarity regarding the reliability analyses. The inclusion of scan-rescan reliability analysis conducted over the two experimental sessions seems counterintuitive given the hypothesis that the DTI metrics would change rapidly with reading training. In other words, if the results had shown significant training-related changes in DTI measures, the reliability tests would not have shown strong scan-rescan reliability. I suggest further justification and discussion of the reliability analysis, or restructuring to include the reliability analysis in the supplementary materials to show the stability of the DTI metrics over time, rather than as a preliminary analysis. Otherwise, please clarify if I am misunderstanding the implementation of the reliability analysis in this study.

Results

- Please clarify the direction of change in left ILF FA that is potentially associated with Alphabet Knowledge improvement. The statement that “BFs signaled a moderate support for the presence of a link between Alphabet knowledge improvement and an FA reduction of the left ILF (BF>3).” (P. 23, lines 392-394) seems to indicate that children who improved more on Alphabet knowledge showed a decrease in FA of the left ILF over time, but this does not seem to be consistent with the explanation in the text or the response to Reviewer 2, in which it seems that the negative correlation between Alphabet Knowledge change and FA change indicates that children with greater Alphabet Knowledge improvement showed smaller increases in FA. It would be helpful to include Figure S3 in the main text to support interpretation of these results.

Reviewer #2: The authors have made a commendable effort in addressing the feedback provided by the reviewers. I have minor feedback regarding the description of the behavioral measures: instead of combining everything into one paragraph, I suggest listing the measures separately for greater clarity and readability.

7. PLOS authors have the option to publish the peer review history of their article (what does this mean?). If published, this will include your full peer review and any attached files.

Reviewer #1: No

Reviewer #2: **Yes: **Silvia Clement-Lam

---

## [Author Response · Author response to Decision Letter 2]

23 Jan 2025

Reviewers' comments:

Have the authors made all data underlying the findings in their manuscript fully available?

Reviewer #1: No

Reviewer #2: Yes

We made sure that the openeuro link works. The full dataset is available here: PREK. OpenNeuro. [Dataset] doi: 10.18112/openneuro.ds005572.v1.0.0

Here is a review link where the dataset can be accessed anonymously: https://openneuro.org/crn/reviewer/eyJhbGciOiJIUzI1NiIsInR5cCI6IkpXVCJ9.eyJzdWIiOiI[…]IjoxNzY4NDAxNzYyfQ.oaZHx1LDOZcFyh0qHt5b-lU1Y-pVZooV-HnYVuvXq5U

Reviewer #1:

The reviewers have carefully addressed all of my initial queries. I have several small suggestions pertaining to the revised version of the manuscript:

1. Data Availability: I attempted to access the data at doi: 10.18112/openneuro.ds005572.v1.0.0 and received the error “403: You do not have access to this page, you may need to sign in.” I was still unable to access the data after signing in with my ORCID.

We made sure that the link works.

The full dataset is available here: PREK. OpenNeuro. [Dataset] doi: 10.18112/openneuro.ds005572.v1.0.0

2. Table 2.

- Typo: Post MRI-camp time different (days)

- Please indicate whether Raw or Standard/Scaled scores are reported (also for the PPVT inclusion criteria, p.8 l. 147).

We added these corrections to Table 2 and page 8 and we clarified that raw scores are reported.

3. Methods

- Please cite software and packages used for statistical analysis.

Statistical analyses were performed using python v3.11 and the following modules: scipy.stats v.1.15.0, statsmodels v0.14.4. We added this information to pages 14 and 17.

4. Thank you for providing additional clarity regarding the reliability analyses. The inclusion of scan-rescan reliability analysis conducted over the two experimental sessions seems counterintuitive given the hypothesis that the DTI metrics would change rapidly with reading training. In other words, if the results had shown significant training-related changes in DTI measures, the reliability tests would not have shown strong scan-rescan reliability. I suggest further justification and discussion of the reliability analysis, or restructuring to include the reliability analysis in the supplementary materials to show the stability of the DTI metrics over time, rather than as a preliminary analysis. Otherwise, please clarify if I am misunderstanding the implementation of the reliability analysis in this study.

High scan-rescan reliability does not necessarily exclude the presence of training-induced DTI changes. This is because Pearson correlation mainly checks if the overall relationship among participants (in the case of subject reliability) or tract nodes (in the case of profile reliability) is constant between the two experimental sessions but it’s not sensitive to changes in the scale of the variables. For instance, we did not expect that reading training would drastically change the shape of the tract profile (i.e. relationship among nodes). Instead we expected a general increase of FA while maintaining the overall shape of the tract profile constant. The presence of a shift in FA values between sessions is still compatible with high profile reliability, as Pearson correlations are invariant to changes in scale. We added a clarification on this point to page 18.

5. Results

- Please clarify the direction of change in left ILF FA that is potentially associated with Alphabet Knowledge improvement. The statement that “BFs signaled a moderate support for the presence of a link between Alphabet knowledge improvement and an FA reduction of the left ILF (BF>3).” (P. 23, lines 392-394) seems to indicate that children who improved more on Alphabet knowledge showed a decrease in FA of the left ILF over time, but this does not seem to be consistent with the explanation in the text or the response to Reviewer 2, in which it seems that the negative correlation between Alphabet Knowledge change and FA change indicates that children with greater Alphabet Knowledge improvement showed smaller increases in FA. It would be helpful to include Figure S3 in the main text to support interpretation of these results.

The reviewer is correct in the interpretation of this correlation, which is surprisingly going in the opposite direction as compared to what we were expecting. We think that children that showed the smaller change in alphabet knowledge are actually those kids that had higher alphabet knowledge at the pre-training session. Hence, they did not have a lot of margin for improvement. What we are observing in this correlation might just be due to a ceiling effect in the alphabet knowledge scale. We updated the paragraph on page 24 to make this point clearer. We do not think that Figure S3 will help better understand the correlations in Table 5 as it does not include information about individual behavioral scores. Hence, we decided to leave it in the Supplementary materials.

Reviewer #2:

The authors have made a commendable effort in addressing the feedback provided by the reviewers. I have minor feedback regarding the description of the behavioral measures: instead of combining everything into one paragraph, I suggest listing the measures separately for greater clarity and readability.

We changed the format of the paragraph on pages 12-14 following this suggestion.

---

## [Editor Report · Decision Letter 2]

26 Jan 2025

Assessing white matter plasticity in a randomized controlled trial of early literacy training in preschoolers

PONE-D-24-32615R2

Dear Dr. Caffarra,

We’re pleased to inform you that your manuscript has been judged scientifically suitable for publication and will be formally accepted for publication once it meets all outstanding technical requirements.

Kind regards,

Signe Bray

Academic Editor

PLOS ONE
---

## [Editor Report · Acceptance letter]

PONE-D-24-32615R2

PLOS ONE

Dear Dr. Caffarra,

I'm pleased to inform you that your manuscript has been deemed suitable for publication in PLOS ONE. Congratulations! Your manuscript is now being handed over to our production team.

Kind regards,

on behalf of

Dr. Signe Bray

Academic Editor

PLOS ONE